# Comparison of Prediction Models Applied in Economic Recession and Expansion

**Dagmar Camska ***  **and Jiri Klecka**

Department of Economics and Management, University of Chemistry and Technology Prague, Technická 5, 166 28 Prague 6, Czech Republic; jiri.klecka@vscht.cz

**\*** Correspondence: dagmar.camska@email.cz

**Abstract:** As a rule, the economy regularly undergoes various phases, from a recession up to expansion. This paper is focused on models predicting corporate financial distress. Its aim is to analyze impact of individual phases of the economic cycle on final scores of the prediction models. The prediction models may be used for quick, inexpensive evaluation of a corporate financial situation leading to business risk mitigation. The research conducted is drawn from accounting data extracted from the prepaid corporate database, Albertina. The carried-out analysis also highlights and examines industry specifics; therefore, three industry branches are under examination. Enterprises falling under Manufacture of metal products, Machinery, and Construction are categorized into insolvent and healthy entities. In this study, 18 models are selected and then applied to the business data describing recession and expansion. The final scores achieved are summarized by the main descriptive statistics, such as mean, median, and trimmed mean, followed by the absolute difference comparing expansion and recession. The results confirm the expectations, assuming that final scores with higher values describe better corporate financial standing during the expansion phase. Similar results are achieved for both healthy and insolvent enterprises. The paper highlights exceptions and offers possible interpretations. As a conclusion, it is recommended that users need to respect the current phase of the economic cycle when interpreting particular results of the prediction models.

**Keywords:** models predicting financial distress; phases of economic cycle; Czech Republic

## 1. Introduction

Forecasting corporate bankruptcy is a crucial task for modern risk management. The current economic environment shaped by globalization, turbulent economic changes, and fierce competition impose challenging conditions for businesses and their prosperity. Contrariwise, many enterprises do not survive in the long run, and they have to withdraw from the market. The findings of the European Commission (2012) show that almost half of new companies went bankrupt within the first five years of their existence. Although corporate defaults seem natural in a market economy, corporate failures have enormous consequences for whole economic systems (Peng et al. 2010; Lee et al. 2011). The consequences can be recognized not only on the macroeconomic but also on the microeconomic level. The parties affected could be suppliers, customers, managers, employees, investors, governmental bodies, and financial creditors. All of these entities want to mitigate business risks and protect themselves from entering or continuing business activities with potentially default entities.

Prediction of corporate bankruptcy or corporate default has been a significant research issue since the 1960s. Pioneering works were associated with names such as Altman (1968) or Beaver (1966). These efforts have led to the construction of prediction models (also called bankruptcy models or models predicting financial distress). These models provide a controlled description of a particular economic reality. It should not be neglected that these models are never 100% accurate as they

work on probability roots based on empirical observations (De Laurentis et al. 2010). The most popular statistical techniques applied are multivariate discriminant analysis and logistic regression (Balcaen and Ooghe 2006; Ohlson 1980). Since 2000, statistical methods have been replaced by artificial intelligence and machine learning methods. These current approaches include neural networks, genetic algorithms, fuzzy logic, vector support machines, or ensemble classifier methods (Alaka et al. 2018; Kumar and Ravi 2007; Lessmann et al. 2015; Acosta-González and Fernández-Rodríguez 2014; Ahn et al. 2000; Du Jardin 2018; Lensberg et al. 2006; Min and Lee 2005; Ravisankar and Ravi 2010; Wu et al. 2010).

De Laurentis et al. (2010) point out that prediction models are part of a broader framework: their limits have to be perfectly understood, and their general application should be avoided. The current modelling approaches make it difficult to fulfill the conditions mentioned above. They do not follow the recommendation by Zellner (1992) known as the KISS principle: Keep It Sophisticatedly Simple, which is often paraphrased as Keep It Simple Stupid. Large financial providers of different types can use the most up-to-date techniques, but credit risk management of small- and medium-sized enterprises (SMEs) differs (Belás et al. 2018).

This different approach used by SMEs causes the popularity of basic statistical techniques to remain unchanged in daily practice and a number of scientific papers can be found as well. Prusak (2018) or Klieštik et al. (2018) provide an overview of the research conducted in selected central and eastern European countries. Research carried out in the area of the Czech Republic involves the works by Karas and Režňáková (2013), Klečka and Scholleová (2010), Čámská (2015, 2016), Machek (2014), and Pitrová (2011). Despite the simplicity, the models predicting financial distress should not be used as dogmas. Two issues discussed in detail within the literature review need to be taken into account. The first is the influence attributed to the economic cycle phases. The second aspect that needs to be considered is the sensitivity of belonging to particular industry sectors. The paper's main aim is to analyze the impact of the economic cycle phases upon final values of the models predicting financial distress, as designed by statistical techniques. The principal conclusions lead to a recommendation that while applying models predicting financial distress, the present current phase of the economic cycle should be respected without regard to a particular industry branch and general corporate financial standing.

This paper has a standard structure and consists of five parts. Section 1 sets the research into a broader context. It describes the terms of the business environment, consequences of corporate defaults, reviews of the current research in this respective field, and explains the paper's main goal. Financial distress and the financially healthy position of a company is defined in Section 2; specific issues, such as the influence of the economic cycle and the role of particular industry sectors are also to be found in this section. Section 3 focuses on the materials and methods, explaining the extraction of the data sample and models predicting financial distress applied. Finally, Sections 4 and 5 present the results of the analysis along with their interpretation, summary, conclusions, and recommendations.

## 2. Literature Review

The review on the sensitivity of the economic cycles can be considered to be helpful for readers to gain an insight into this research. The theoretical background also refers to some other issues related to models predicting financial distress. The first defines financial standing, considering healthy and distressed. It is necessary to classify companies correctly before prediction models are applied. Secondly, the companies under investigation must be assigned into relevant industry sectors. The type of industry influences the risk of bankruptcy, sensitivity to the economic cycles, and, particularly, the values of financial ratios entering into prediction models. The models applied will be discussed separately in Section 3.

Deterioration of the overall economic situation results in an increased number of bankruptcies (Svobodová 2013; Achim et al. 2012; Smrčka et al. 2013). Bruneau et al. (2012) examined whether corporate bankruptcies are influenced by macroeconomic variables and whether defaults determine

the business cycle in France. Altman (2004) emphasized the impact of a turbulent economic environment on an increasing unexpected number of bankruptcies in the United States in 2001 and 2002. Liou and Smith (2007) considered including macroeconomic variables into prediction models as a logical step but also admitted that it happens only very rarely. Several other studies confirm that the use of macroeconomic variables improves the predictive accuracy of models (Korol and Korodi 2010; Hol 2007; Zhou et al. 2010). The main drawback of these approaches applied is that only one economic period is scrutinized and comparison over time is missing. Surprisingly, Topaloglu (2012) is an exception because the paper covers American bankruptcies in the manufacturing industry during the period 1980–2007, which allows the conclusion that accounting variables lose predicting ability when market-driven variables are included.

Macroeconomic deterioration triggers the increase of corporate defaults and it probably also influences values of financial ratios, which would result in the changed final values of models predicting financial distress. During the recession phase, the values of economic indicators could be expected to deteriorate contrary to the phase of expansion when these values would get improved. The question arises whether the impact described is significant and observable in most economic indicators, entering into the models predicting financial distress. Li and Faff (2019) concluded that market-based information assumes importance during periods of financial crisis, in contrast to accounting-based information, the importance of which in the same phase is reduced. It seems that bankruptcy models based on macroeconomic variables are not stable over time since they are not used recurrently, and neither are they scrutinized in the longer time horizons. It seems that the life cycle of prediction models containing macroeconomic variables is not long enough and cannot be used for more economic cycles.

To achieve the required accuracy in model testing, it is essential to categorize the enterprises correctly; basically, into one of two main groups, either as healthy or distressed entities. Financial distress can be defined in many different ways, and similarly, the terminology referring to such companies also differs (bankrupt, insolvent, in default). Merton (1974) defines the default as a situation when the enterprise value is lower than the value of debts. Moyer (2005) compares corporate financial distress to the situation when the box of assets becomes smaller than the box of debts. Using this approach, the enterprises are distinguished through their over indebtedness, such as in Schönfeld et al. (2018). Insolvency is mostly connected with the inability to pay debts, which can be short or long-termed (Crone and Finlay 2012; Deakin 1972; Du Jardin 2017; Foster 1986). Another possibility to define a default is a definition provided by credit rating agencies. The approach used by Moody's can be found in Hamilton et al. (2001). For research purposes, data availability has to be respected. Some research works are based on non-public information provided by financial creditors. In this research, however, only publicly available information is used exclusively. As a result, financial default is defined as corporate insolvency under the Czech Insolvency Act (Act No. 182/2006 Coll.). Insolvency can be declared because of an inability to pay claims or because of over indebtedness. The second group of companies examined is presented as healthy companies. Less attention is given to the definition of healthy enterprises in literature. In this paper, healthy companies are considered to be those having positive economic value added (Jordan et al. 2011). This approach was applied in Čámská (2015, 2016).

The last issue covered in this literature review is the sensitivity of belonging to a particular industry branch. Ganguin and Bilardello (2004) point out that some industries are riskier than others. They conclude that the type of industry influences the risk of deterioration. One reason for industry sensitivity is its exposure to the risk of default. Another reason is that different industries achieve different performance. The literature provides numerous pieces of evidence for this statement. Structure of capital sources (proportion of equity and liabilities) is determined by belonging to industry sectors (Frank and Goyal 2009; Öztekin 2015). Structure of working capital and connected corporate liquidity are also influenced by industries (Vlachý 2018). The same can be said about corporate profitability (Jackson et al. 2018). Belonging to a particular industry sector has an impact on the quality of financial performance predicted (Fairfield et al. 2009; Lee and Alnahedh 2016). Chava and Jarrow (2004) even

highlighted that the coefficients of the models predicting financial distress should be calibrated according to the particular industry branches. This leads to a conclusion that financial ratios influenced by industry specifics entering into bankruptcy models could influence the results achieved. Due to these reasons, this paper strictly separates individual industry branches.

## 3. Materials and Methods

This part describes the materials and methods employed herein. The materials include the observations extracted from the prepaid corporate database, Albertina. The selected observations all have to meet some predefined requirements. Each observation describes one company and is based on the annual financial statements. The methods specify the steps conducted during the analysis leading to the results achieved. The description provided below contains a sufficient number of details, therefore allowing any professionals to replicate this research work.

### 3.1. Materials

This paper's idea is verified by the data specified in this subchapter. Information about corporate financial performance is mainly included in financial statements, such as a balance sheet and income statement. This quantitative research includes hundreds of companies, which means that the data analysis is based on publicly available financial statements. The selected financial statements were extracted from the prepaid corporate database, Albertina. What proved to be the main obstacle was rather complicated access to data since many companies do not report on time or they tend not to report at all despite reporting being an obligatory legal requirement in the Czech Republic. Some further details concerning the Czech disclosure discipline can be found in Strouhal et al. (2014) focusing on TOP100 companies according their sales and in Bokšová and Randáková (2013) focusing on insolvent entities.

The data selected and obtained can be divided into several subcategories. The first category includes data strictly polarized; on one hand, there are enterprises which declared insolvency. On the other hand, there are companies considered financially healthy due to their positive economic value added creation (Jordan et al. 2011). Their return on equity exceeds the required level of return published by the Ministry of Industry and Trade (2013, 2018). Both groups can be divided into two subparts describing different time periods. There are the companies which announced insolvency as a consequence of the latest global economic crisis in 2012 and 2013, and businesses which announced their insolvency after the year 2014 until the first quarter of 2019 during economic expansion. The analyzed financial statements always describe the accounting year one or two periods before the companies had become insolvent. The same process was applied to the healthy entities. The preceding sample focuses on the accounting year 2012 and the current one describes the year 2017. The year 2017 was selected for this research for the following reasons. The financial data for the year 2019 have not been reported yet and neither those for the year 2018 have been published in full. Secondly, the data sample contains three industry branches, specifically, Manufacture of fabricated metal products, except machinery and equipment (CZ-NACE 25), Manufacture of machinery and equipment (CZ-NACE 28), and Construction (CZ-NACE F). Previous works mentioned in the literature review confirmed that industry specifics are relevant. The companies in this research, therefore, needed to be classified according to their industry sectors. These sectors provide one of the largest homogenous data samples, i.e., for the purposes of this research they were not selected randomly.

Table 1 shows the structure of the data sample following the aforementioned description. Healthy and insolvent enterprises are strictly polarized. The years 2012 and 2017 reflect different periods for comparison. According to the economic cycle, the year 2012 represents a recession phase and the year 2017 an economic expansion phase. Special emphasis should be placed on the analyzed industry sectors—CZ-NACE 25, CZ-NACE 28, and CZ-NACE F. It seems that a particular phase of the economic cycle influenced the number of the businesses extracted from the Albertina database, confirming the logical premises of economic cycles in general. Significantly, healthy companies can be observed more

frequently in the expansion phase, however, insolvent enterprises can be found more frequently in the recession phase. This also explains why the second time period for extracting insolvent enterprises cannot be shorter. The sample size for the insolvent companies would be negligible if the period was shortened. The only exception observed is the number of insolvent entities within the construction industry during the expansion period. This number is three times larger than during the recession period. This can be explained by the ongoing construction sector crisis or better disclosure discipline.

**Table 1.** Size of the analyzed sample.

| Industry Branch | Healthy 2012 | Insolvent 2012 | Healthy 2017 | Insolvent 2017 |
|---|---|---|---|---|
| CZ-NACE 25 | 383 | 36 | 786 | 25 |
| CZ-NACE 28 | 33 | 10 | 321 | 11 |
| CZ-NACE F | 229 | 33 | 1997 | 105 |

Source: authors' own work.

*3.2. Methods*

The analysis carried out was based on models predicting financial distress, whose accuracy was confirmed and verified on Czech businesses in previous works (Čámská 2015, 2016). Methods such as linear discriminant analysis and logistic regression belong to classical statistical methods applied in prediction of corporate default risk. The models applied were designed using these statistical techniques. Their frequent reuse depends on their ease of use and clear interpretability. Users do not need to have deep insight into advanced statistical, as well as non-statistical, techniques.

The conducted analysis was based on the 18 following models predicting financial distress. The bankruptcy models in this paper are marked by the following numbers: 1—Altman, 2—IN01, 3—IN05, 4—Doucha, 5—Kralicek, 6—Bonita, 7—Prusak 1, 8—Prusak 2, 9—PAN-E, 10—PAN-F, 11—PAN_G, 12—D2, 13—D3, 14—Hajdu and Virág, 15—Šorins and Voronova, 16—Merkevicius, 17—R model, and 18—Taffler. The exact models' specifications are accessible in the relevant literature cited below. The models introduced were designed in different countries and in different periods. Some models were constructed in the most developed economies and at the beginning of 1990s, were assumed to be best practice in the Czech Republic. In the late 1990s, these foreign designs were replaced by domestically designed models. These efforts were visible not only in the Czech Republic, but also in other countries in the Central and Eastern European region. Countries like Poland, Hungary, Lithuania, and Latvia, due to historical circumstances, underwent similar political and economic development as the Czech Republic.

The approaches imported from the most developed economies are represented by the American Altman Z-Score formula (Altman 1993), German Bonita Index (from the German original Bonitätsanalyse) (Wöber and Siebenlist 2009), Austrian Kralicek Quick Test (Kralicek 2007), and British Taffler (Agarwal and Taffler 2007). National efforts from previously transitioned economies described in this research include Czech IN01 (Neumaierová and Neumaier 2002), IN05 (Neumaierová and Neumaier 2005), Balance Analysis System by Rudolf Doucha (Doucha 1996), Polish Prusak 1, Prusak 2, PAN-E, PAN-F, PAN-G, D2, D3 (all described in Kisielińska and Waszkowski 2010), Hungarian Hajdu and Virág (Hajdu and Virág 2001), and Baltic approaches, such as Šorins and Voronova (Jansone et al. 2010), Merkevicius (Merkevicius et al. 2006), and R model (Davidova 1999).

The conducted analysis was then divided into the following phases. At the beginning, final values of the aforementioned models predicting financial distress were calculated for individual companies included in the data sample. Then, the final values calculated were summarized. Their summary was performed by general descriptive statistics, such as mean, median, or trimmed mean. Finally, the comparison between the time period of expansion and recession was conducted. The time of expansion was represented by the data sample describing the year 2017 defined previously. In contrast, the time

of recession was presented by the data sample of the year 2012. The comparison was based on absolute differences expressed by Equations (1) and (2).

$$\text{Absolute difference} = \text{Indicator value}_{2017} - \text{Indicator value}_{2012}, \tag{1}$$

$$\text{Absolute difference} = (\text{Indicator value2017} - \text{Indicator value2012}) \times (-1). \tag{2}$$

The first equation was applied to 17 tested models whose higher values mean better financial standing. The second equation was applied to one model only. This exception is the Kralicek Quick Test (marked by the number 5) which has an opposite metric. Better financial standing is connected with a lower, not higher, final value. This explains why other forms to express the absolute difference were used. As for the two differences displayed above, their positive value reflects a more favorable classification of companies in 2017 and a negative value indicates a more favorable classification of companies in 2012.

## 4. Results

This part is dedicated to the results achieved. The main aim of this study was to examine the difference in corporate financial standing during a recession and expansion phase of the economic cycle. It also emphasizes the sensitivity of industry sectors and general differences in financial standing (healthy contrary to insolvent companies). The results are most frequently demonstrated by their visualization, as proposed by Čámská (2019). This process was chosen as a number of models were employed and it highlights the differences between the industry branches.

Statistical characteristics, such as the mean, median, and trimmed mean, were calculated for each subsample. Table 2 shows an example of the results when applying the Altman model for the insolvent and healthy enterprises in 2017. It seems that from a statistical point of view, some enterprises could serve as outliers. Since these entities represent realistic financial standing, the question whether to exclude them from the sample can be considered rather controversial. The healthy group contains mainly positive outliers whose financial standing is significantly better. In contrast, the insolvent group mostly consists of negative outliers whose financial standing is considerably worse. This affects the mean value. The trimmed mean cannot rely on the same assumption due to the different sample sizes. For healthy enterprises, the mean limitation of 1/20 (5%), except CZ-NACE 28 in 2012 (which uses 1/10 (10%)), was applied. The situation is much more difficult in the case of insolvent companies. The mean limitation of 1/10 (10%), except CZ-NACE 28 in 2012 and also in 2017 (in these cases applied limitation of 1/5 (20%)), was used. This suggests that the median is an optimal indicator for visualization.

**Table 2.** Descriptive statistics for Altman's Z-score in 2017.

| Statistics | Healthy | Healthy | Healthy | Insolvent | Insolvent | Insolvent |
| --- | --- | --- | --- | --- | --- | --- |
| | Mean | Median | Trimmed Mean | Mean | Median | Trimmed Mean |
| CZ-NACE 25 | 4.17 | 3.71 | 3.98 | −1.23 | 0.74 | −0.04 |
| CZ-NACE 28 | 4.27 | 3.70 | 4.09 | 1.55 | 1.68 | 1.52 |
| CZ-NACE F | 4.22 | 3.73 | 4.04 | 0.72 | 0.83 | 0.76 |

Source: authors' own work.

Visualization can also express several criteria. The first takes into account a type of time period. In our case, the periods monitored (2012 and 2017) are different. The second criterion takes into consideration the type of industry branches. Again, the companies selected for the purpose of this study belong to three different industry branches. The third criterion applied here distinguishes companies according to their financial standing. The companies surveyed herein differ in their financial standing, presenting a strict polarization. Figures 1 and 2 display the results for the industry sector CZ-NACE F Construction. Figure 1 demonstrates healthy entities contrary to the insolvent companies presented in Figure 2. The different phases of the economic cycle are displayed by the separated curves in each figure. Models 5 and 18 (Kralicek and Taffler) are not included in the final visualization. It has

been already highlighted that the Kralicek model is based on a different metric system, which leads to different results from the other applied models. The Taffler model has also been excluded due to its values range exceeding other prediction tools by 2–3 times. Higher total values are caused by the used individual indicators and especially assigned weights, which were chosen during the model's design.

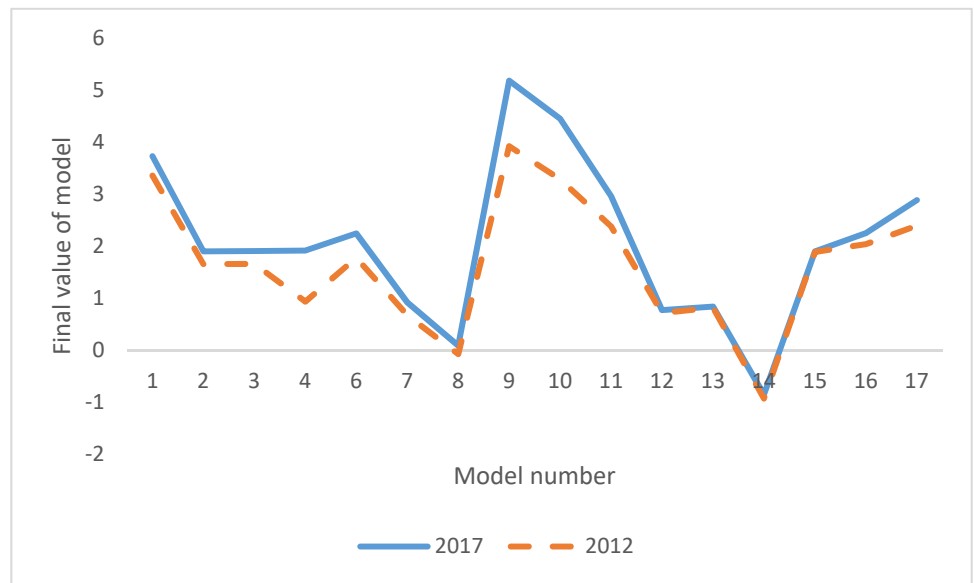

**Figure 1.** Indicator values of the studied models for healthy companies in Construction (CZ-NACE F). Source: authors' own work.

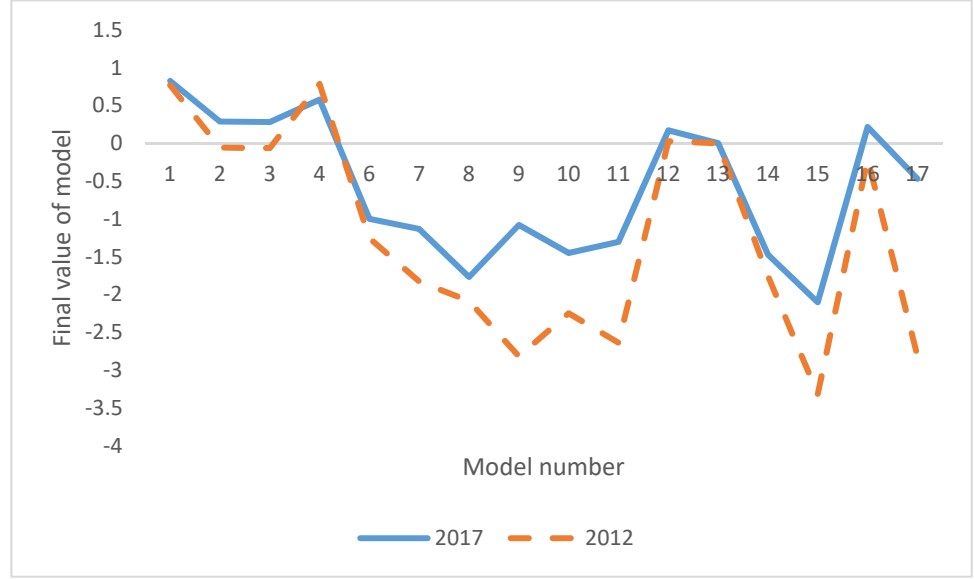

**Figure 2.** Indicator values of the studied models for insolvent companies in Construction (CZ-NACE F). Source: authors' own work.

Figure 1 confirms the research hypothesis that the recession phase leads to lower final values of models predicting financial distress. Figure 1 works with the companies defined as financially healthy. Figure 2 provides additional support for this claim and the differences for insolvent companies are even more significant. In the case of the models marked as 12 and 13, it should be noted that they were constructed by logistic regression and therefore their range of final values is from 0 to 1. Their visualized differences are insignificant in comparison with other models designed by linear

discriminant analysis. The changed scale of graph would reveal that the differences are observable also for the models based on logistic regression.

Both Figures 1 and 2 concentrate on just one particular industry branch. They demonstrate the differences of the economic phases for the economic activity of CZ-NACE F Construction. The need to differentiate between sectors has already been emphasized in the theoretical part. The literature review highlighted the sensitivity of models predicting financial distress to particular industry branches. Figures 3 and 4 display results achieved in all branches included in the sample. The sample includes not only Construction (CZ-NACE F), but also Manufacture of fabricated metal products, except machinery and equipment (CZ-NACE 25), and Manufacture of machinery and equipment (CZ-NACE 28).

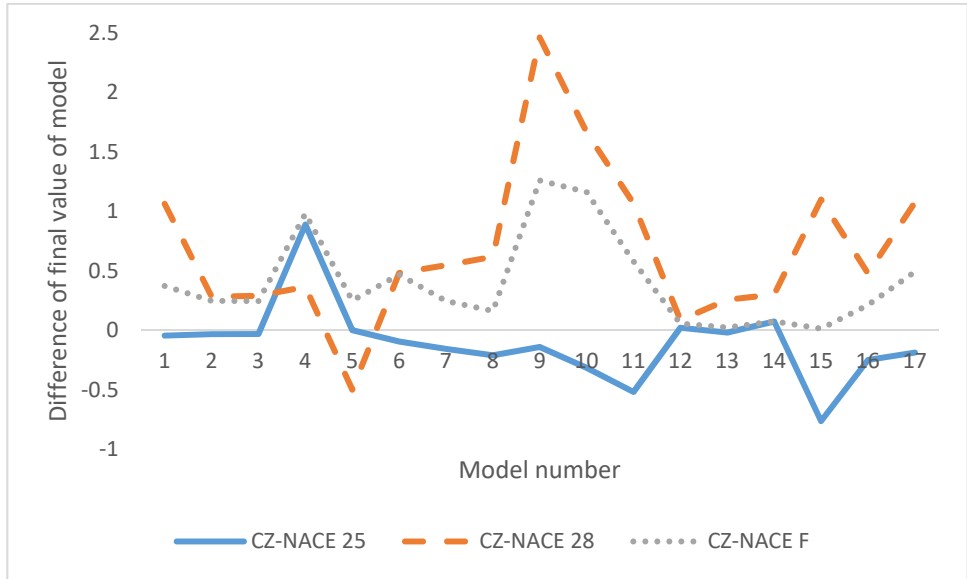

**Figure 3.** Absolute differences of the model indicators for healthy companies regarding 2012 and 2017 by branches. Source: authors' own work.

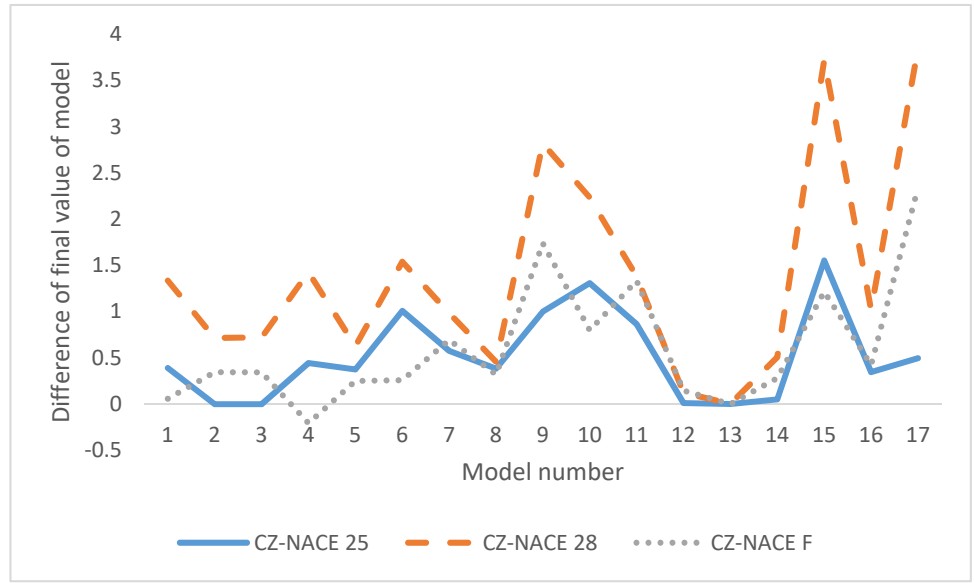

**Figure 4.** Absolute differences of the model indicators for insolvent companies regarding 2012 and 2017 by branches. Source: authors' own work.

Table 3 displays the results of the Wilcoxon test applied to all three industry branches studied. The null hypothesis is that there is no difference between the year 2012 and the year 2017. Small

p-values lead to the rejection of the null hypothesis and to the acceptance of alternatives. The alternative can be presented as there are differences in the indicator values of the tested models between the recession phase (2012) and the expansion one (2017). The analysis was conducted for the healthy and also insolvent enterprises. *p*-values smaller than 10% are highlighted in the table. In these cases, the null hypothesis was rejected and the alternative was accepted. CZ-NACE 28 (Machinery) and CZ-NACE F (Construction) have reached convincing results for most models in the healthy, and also insolvent, sample. CZ-NACE 25 (Manufacture of metal products) does not support the alternative in many cases.

**Table 3.** *p*-values of the Wilcoxon test.

| Company Type | Healthy | Healthy | Healthy | Insolvent | Insolvent | Insolvent |
|---|---|---|---|---|---|---|
| Model | CZ-NACE 25 | CZ-NACE 28 | CZ-NACE F | CZ-NACE 25 | CZ-NACE 28 | CZ-NACE F |
| Model 1 | 0.1246 | 0.0094 | 0.0177 | 0.3556 | 0.0167 | 0.4495 |
| Model 2 | 0.9158 | 0.0219 | 0.0003 | 0.2126 | 0.1213 | 0.2582 |
| Model 3 | 0.9118 | 0.0220 | 0.0003 | 0.2072 | 0.1213 | 0.2541 |
| Model 4 | 0.0000 | 0.0039 | 0.0000 | 0.2467 | 0.2908 | 0.4406 |
| Model 5 | 0.0219 | 0.2928 | 0.0001 | 0.0455 | 0.1329 | 0.0390 |
| Model 6 | 0.8835 | 0.1783 | 0.0002 | 0.1236 | 0.0573 | 0.0666 |
| Model 7 | 0.3279 | 0.0582 | 0.0921 | 0.1347 | 0.0573 | 0.0768 |
| Model 8 | 0.1380 | 0.0418 | 0.2294 | 0.0889 | 0.1213 | 0.1336 |
| Model 9 | 0.4733 | 0.0083 | 0.0037 | 0.1309 | 0.0039 | 0.0811 |
| Model 10 | 0.4938 | 0.0361 | 0.0007 | 0.1549 | 0.0290 | 0.0864 |
| Model 11 | 0.2264 | 0.1174 | 0.0231 | 0.1347 | 0.0573 | 0.0156 |
| Model 12 | 0.1269 | 0.0070 | 0.0002 | 0.2180 | 0.2599 | 0.3074 |
| Model 13 | 0.9997 | 0.0262 | 0.3491 | 0.5379 | 0.1392 | 0.6407 |
| Model 14 | 0.9026 | 0.0926 | 0.5013 | 0.9415 | 0.9439 | 0.4031 |
| Model 15 | 0.0000 | 0.0465 | 0.5648 | 0.2910 | 0.0112 | 0.0245 |
| Model 16 | 0.0084 | 0.0440 | 0.2563 | 0.2292 | 0.0060 | 0.0673 |
| Model 17 | 0.4545 | 0.0397 | 0.0635 | 0.2292 | 0.0137 | 0.0029 |
| Model 18 | 0.5437 | 0.1081 | 0.1799 | 0.2778 | 0.0167 | 0.0158 |

Source: authors' own work.

Figures 3 and 4 do not reflect any distinction between the economic phases as their curves show the absolute differences defined by Equations (1) and (2). This enables the inclusion also of the Kralicek Quick Test into the graphs. Taffler, however, remains excluded and will be presented in a separate table. The curves above the horizontal axis mean that the models predicting financial distress reached higher values for the economic phase of expansion. On the contrary, curves below the horizontal axis mean that the bankruptcy models had higher values during the economic phase of recession.

Figure 3 represents healthy enterprises. The results obtained in the sectors of Construction and Machinery confirm the expectations. Final values of models predicting financial distress were all higher in the expansion phase except for the Kralicek Quick Test in the case of CZ-NACE 28. Surprisingly, CZ-NACE 25 (Manufacture of fabricated metal products) did not meet the expectations of the conducted research. The blue curve is situated below the horizontal axis for most models. It means that most models predicting financial distress provided better results for the recession than for the expansion phase in the case of CZ-NACE 25. The reasons will be explained below in the discussion.

The results of healthy companies are followed by the results for insolvent enterprises displayed in Figure 4. There are no significant differences between the individual industry branches subjected to analysis. The curves are situated above the horizontal axis, except for the Doucha approach, which was applied to the Construction sector. The results achieved can interpret the financial situation of the insolvent companies as significantly worse in the recession phase or significantly better in the expansion phase of the economic cycle, which met the preliminary expectations.

Friedman's test for comparing model performances for the different branches was applied. The results of the test provide the following interpretation. In the case of healthy companies, the industry branches CZ-NACE 28 (Machinery) and CZ-NACE F (Construction) do not differ significantly. In contrast, the industry sectors CZ-NACE 25 (Manufacture of metal products) and CZ-NACE F (Construction), as well as the pair CZ-NACE 25 (Manufacture of metal products) and CZ-NACE 28 (Machinery), differ significantly. The interpretation in the case of insolvent companies is following. The pairs CZ-NACE 28 (Machinery) + CZ-NACE F (Construction) and CZ-NACE 25 (Manufacture of metal products) + CZ-NACE 28 (Machinery) differ significantly. On the other hand, the industry sectors CZ-NACE 25 (Manufacture of metal products) and CZ-NACE F (Construction) do not differ significantly.

Again, the Taffler model has been excluded from the visualization. Its results are presented separately and can be seen in Table 4. Taffler's absolute difference in the median confirms previous outcomes. The prediction models for the healthy enterprises belonging to the sector of Manufacture of metal products showed better scores in the recession phase (leading to the negative value of absolute differences). Other industry sectors, with no respect for basic financial standing, show positive values, which can be interpreted as better financial conditions in the expansion phase in contrast to the recession phase.

**Table 4.** Absolute differences of the Taffler model regarding 2012 and 2017 by branches.

| Company Type | Healthy | Insolvent |
|---|---|---|
| CZ-NACE 25 | −1.03 | 2.47 |
| CZ-NACE 28 | 3.78 | 9.57 |
| CZ-NACE F | 2.08 | 5.94 |

Source: authors' own work.

The visualization submitted in the figures represents results without using in-depth statistical methods. The apparent advantage of this approach is the opportunity for quick interpretation by the user, without requiring in-depth statistical knowledge. Figures 5 and 6 show the summarized results achieved on statistical bases. As demonstrated in a visualization, the expectations failed to be met in all models predicting financial distress applied. Figures 5 and 6 contain results for descriptive statistics, such as the mean, median, and trimmed mean. A number of models confirming expectations (absolute frequency) is followed by the share of models confirming expectations (relative frequency). Models confirming expectations had the curves above the horizontal axis in Figures 3 and 4. Their values were higher in the expansion rather than in the recession period.

Figures 5 and 6 confirm that the selected descriptive statistic for visualization (median versus mean and trimmed mean) does not influence results significantly. The Machinery (CZ-NACE 28) and Construction (CZ-NACE F) sectors provided comparable results for most models, regardless whether companies were healthy or insolvent. As already discussed, the Kralicek Quick Test failed in the field of Machinery in the case of healthy enterprises. The same can be applied to the Šorins–Voronova model in the field of Construction. On the contrary however, the majority of models failed in the case of healthy enterprises belonging to the Manufacture of metal products. Only median analysis based on Doucha, D2, and Hajdu and Virág models reached a satisfactory outcome.

The situation of insolvent entities is displayed in Figure 6. The level of error seems much lower as many models detected insolvency correctly. The lowest accuracy occurs again in the Manufacture of metal products. In the case of median models, such as IN01, IN05, and D3, were against the expectations in CZ-NACE 25. All models applied to CZ-NACE 28 reached expectations. Unconvincing results (mean) were provided by models such as Doucha, Bonita, and Prusak 1 in the field of Construction. The Doucha model collapsed for all three descriptive statistics in this case.

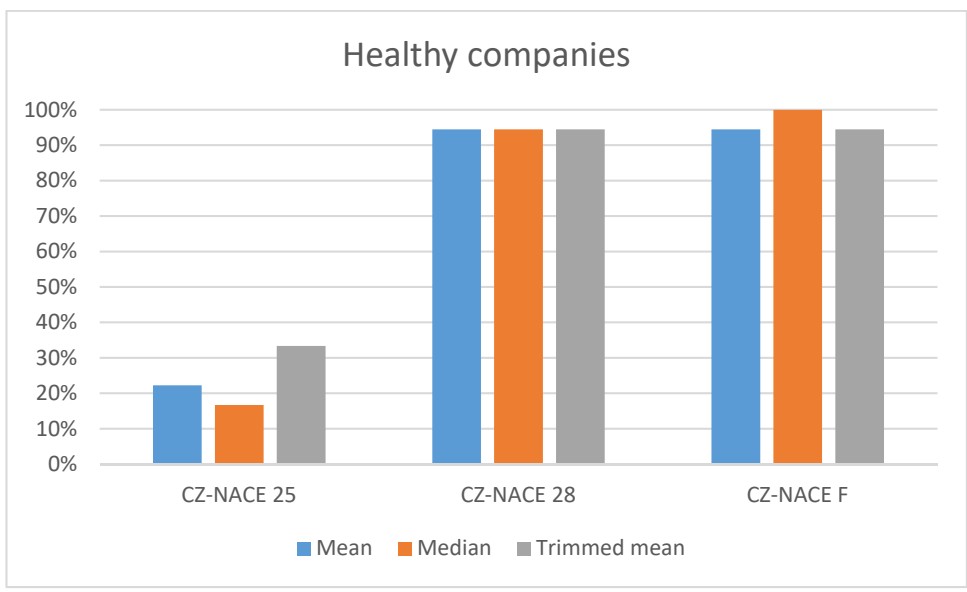

**Figure 5.** The number of models within each branch confirming better conditions for healthy companies. Source: authors' own work.

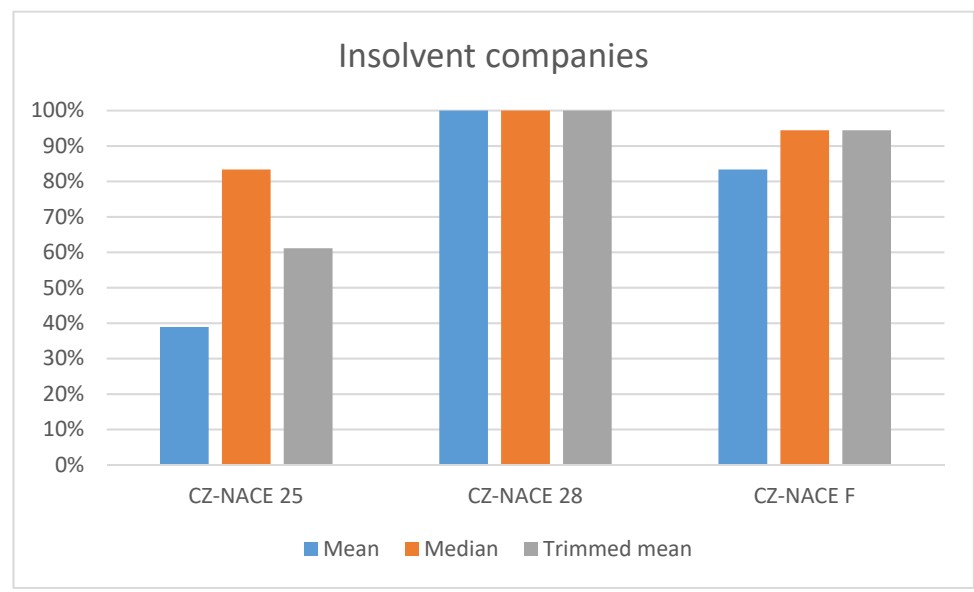

**Figure 6.** The number of models within each branch confirming better conditions for insolvent companies. Source: authors' own work.

## 5. Discussion and Conclusions

The results achieved, as described above, confirmed the working hypothesis that the phase of economic cycle influences corporate financial standing. Worse financial standing is expected in a recession phase and better financial conditions during an expansion phase. This finding has a significant consequence on models predicting financial distress related to forecasting corporate financial situation. If models predicting financial distress are applied, the users should respect overall economic conditions, including macroeconomic and industry development. The recession phase mostly leads to lower final scores of bankruptcy models; on the contrary, the expansion phase leads to higher final scores. The evaluation of a company, according to models predicting financial distress, should take into account the phases of the economic cycle. It seems it is not necessary to include macroeconomic

variables into models, but the overall economic situation should be considered at least in an expert's decision when the final scores are interpreted.

The part describing the results emphasized the issue of healthy companies belonging to CZ-NACE 25. Figure 3 and Table 4 proved that most models predicting financial distress had better results in a recession period. This observation contradicts the expectations and results in other sectors (CZ-NACE 28 and CZ-NACE F). It should also be highlighted that the results of insolvent enterprises fulfilled the expectations. One explanation for this can be as follows. Firstly, the data sample of 2012 was previously extracted in the year 2014 for other research and the methodology applied was slightly different. Healthy companies should have created positive economic value added in three years in a row between the years 2010 and 2012, although only one year of positive economic value added was required for the data sample of 2017. This requirement excluded many companies as they were not deemed entirely financially healthy, but the same was applied to other analyzed sectors. Secondly, the industry situation and its development can influence the results. The development of the Manufacture of metal products (CZ-NACE 25) can be different from the development of Machinery (CZ-NACE 28) and Construction (CZ-NACE F).

Unconvincing results were obtained for different models predicting financial distress in three industry branches under examination. The unconvincing results are not a consequence of the models' design alone. If the users decide to predict financial distress promptly, they should use more than one prediction model. Multiple verifications can eliminate the randomness discussed previously. It is essential to realize that models predicting financial distress are designed for a quick evaluation of the corporate financial situation, they work on empirical bases, and they never function as natural law (De Laurentis et al. 2010). It should also be respected that economics belongs to the social sciences, although many processes can be quantified, and the behavior of economic entities can be described systematically.

Future research directions would benefit from the application of advanced statistical techniques. The methods enabling self-adaptation and learning are likely to have a unique position. Some approaches taking advantage of macroeconomic or industry variables are not published since they are part of the company's know-how. Although large financial providers of different kinds use these techniques, small- and medium-sized enterprises cannot apply them for mitigating their business risk. Unfortunately, as current research directions tend to move away from widespread application in practice, the primary intentions presented by Altman (1968) or Beaver (1966) are not met.

**Author Contributions:** Conceptualization, D.C. and J.K.; methodology, D.C.; validation, J.K.; formal analysis, J.K.; resources, D.C. and J.K.; data curation, J.K.; writing—original draft preparation, D.C.; writing—review and editing, D.C. and J.K.; visualization, J.K.; supervision, J.K.; project administration, D.C.; funding acquisition, D.C. All authors have read and agreed to the published version of the manuscript.

**Funding:** The paper is one of the outputs of the research project "Financial characteristics of enterprise in bankruptcy" registered at Grant Agency of Academic Alliance under the registration No. GAAA 10/2018.

**Conflicts of Interest:** The authors declare no conflict of interest. The funders had no role in the design of the study; in the collection, analyses, or interpretation of data; in the writing of the manuscript, or in the decision to publish the results.

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
