# Peer review of "Comparison of Prediction Models Applied in Economic Recession and Expansion"

_jrfm, doi:10.3390/jrfm13030052_

Round 1

Reviewer 1 Report

This paper applied and compared 17 corporate default prediction models to Czech companies in manufacturing industry. However, I have following questions.

How companies are grouped by healthy and insolvent? Is the grouping based on cash-flow insolvency (liquidity problem) or balance-sheet insolvency (negative net assets)? Is the grouping based on one financial ratio, multiple ratios, or others and what is/are it/they? Whether the grouping method is consistent with the method used by the 18 models applied in this paper? Were all the 18 models built based on the same definition of defaults/insolvency in the original literature?

Author Response

Author would like to thank the reviewers for the valuable questions which are partly answered in the text, specifications are provided further.

Question 1

How companies are grouped by healthy and insolvent?

Response 1

Healthy companies are companies which created possitive economic value added. It means their return on equity achieved exceeded required level of return published by Ministry of Industry and Trade. Sources are provided. The explanation was added into the text.

Healthy and also insolvent companies are grouped according to their belonging to the specified industry branches.

Insolvent companies are the companies which declared officially insolvency. The other details provided in the text.

Healthy and insolvent companies are matched via the related industry sector, the time of observed data. Other factors were not chosen for matching.

Question 2

Is the grouping based on cash-flow insolvency (liquidity problem) or balance-sheet insolvency (negative net assets)?

Response 2

The grouping is based on valid insolvency law in the Czech Republic. Insolvency can be declared because of liquidity problem as well as negative net assets. Information is provided in the text.

Question 3

Is the grouping based on one financial ratio, multiple ratios, or others and what is/are it/they?

Response 3

Answers are already provided in the previous parts.

Question 4

Whether the grouping method is consistent with the method used by the 18 models applied in this paper?

Response 4

It is difficult to provide an answer because the original literature does not provide all the time the description which allows full replication. Insolvent companies are mainly the companies which went bankrupt officially (with respect to law) in the original literature. The healthy companies are hardly definied by the authors.

Question 5

Were all the 18 models built based on the same definition of defaults/insolvency in the original literature?

Response 5

18 original models were based on official bankruptcy/insolvency declaration. The definition does not need to be the same because the models were designed in different countries whose national law can differ. The insolvecy law follows the same sense in all developed countries but there can be some local and time discrepancies which cannot be evaluated by authors.

Reviewer 2 Report

The article compared 18 corporate bankruptcy models applied for enterprises operating in the manufacture industry. I appreciate the paper very much because it is original, interesting and follows a logical structure. However there are many flaws in the article which should be corrected. I support publishing this article after a major correction. The first thing to mention is the language style and grammar which makes it very hard to understand the content of the paper. Just to mention a few serous mistakes:

In line 58-59 the sentence makes no sense: „This different approach of SMEs causes that the popularity of basic statistical techniques remains unchanged in practice and a number of scientific papers can be found as well”.  In line 61: „In the area of the Czech Republic there can found research works”, instead of „can found” should write „can be found” and the style is not correct here (In the area of the Czech Rep.). In line 64: instead of „in detail discussed” authors should write „discussed in detail”. In line 66: instead of „the main paper’s aim” should write „the paper’s main goal”. In line 133-136 the formatting of the text is different. I’m not going to continue with all the grammar mistakes of the article. It should be checked and corrected by a native referee.

In my humble opinion the exception (Kralicek Quick and Taffler model should be discussed in more details in the article, maybe the form of the model should be described).

The most problematic part is the results section. The titles of the figures and tables are not meaningful and it is hard to understand the figures at the first sight. Table 2 should be named as: Descriptive statistics for Altman’s Z score in 2017. Instead of NACE codes authors are better to use names. For example CZ-NACE F is construction, so refer to it as construction.

Figure 1 should be named as „indicator values of the studied models for Healthy companies in Construction”. The x axis should be names as model number, y axis should be named as Z’score or whatever it is. The same holds for Figure 2. Figure 3 is not understandable in its current form. Instead of 25, 28 and F authors should have written CZ-NACE-25/28/F but the names are better here (Construction, Fabricated metals, Machinery and equipment). The figure should be named as „Absolute differences of the model indicators for Healthy companies regarding 2012 and 2017 by branches”. The same holds for Figure 4. The name of Table 4 should be changed to:

„The number of models within each branch confirming better conditions in expansion for Healthy companies„. The same is true for Table 5. In my humble opinion table 4 and 5 should be switched with a Figure and absolute frequency should be omitted and represent only the relative frequencies.

The biggest mistake of the article is that it claims significant differences that are not proved and tested by statistical methods. For example in line 318: „in Figure 4. There are no significant differences between…”. I suggest the authors to run a non-parametric test for two related samples comparing model indicators per branches for years 2012 and 2017 like Wilcoxon test (Figure 1,2) or Friedman’s test for comparing model performances for the different branches (Figure 3,4).

Author Response

Authors would like to thank the reviewers for their valuable comments, suggestions, and recommendations. Authors greatly appreciate their hard work which helped to improve the paper, eliminate discrepancies and misunderstandings that authors did not notice or did not realize. Details are provided bellow.

Comment 1

In line 58-59 the sentence makes no sense: „This different approach of SMEs causes that the popularity of basic statistical techniques remains unchanged in practice and a number of scientific papers can be found as well”.  In line 61: „In the area of the Czech Republic there can found research works”, instead of „can found” should write „can be found” and the style is not correct here (In the area of the Czech Rep.). In line 64: instead of „in detail discussed” authors should write „discussed in detail”. In line 66: instead of „the main paper’s aim” should write „the paper’s main goal”. In line 133-136 the formatting of the text is different. I’m not going to continue with all the grammar mistakes of the article. It should be checked and corrected by a native referee.

Response 1

The text has been rewritten with the help of a language teacher and the final version has been checked by a native speaker.

The incorrect formatting of the text was changed.

Comment 2

In my humble opinion the exception (Kralicek Quick and Taffler model should be discussed in more details in the article, maybe the form of the model should be described).

Response 2

Kralicek Quick Test and Taffler model can be found in the relevant literature provided. Their main idea is similar as for the other models, they do not differ in their aim. Taffler is completely the same although its values are generally much higher and therefore they distort the used visualization. It is caused by the used indicators and especially their assigned weights during the model design. The reason has been added into the text.

The exception of Kralicek Quick Test is explained in the paper (rows 240-244). The recalculated absolute difference enables to display the model in figure 3 and 4 without any difficulties.

Comment 3

The most problematic part is the results section. The titles of the figures and tables are not meaningful and it is hard to understand the figures at the first sight. Table 2 should be named as: Descriptive statistics for Altman’s Z score in 2017. Instead of NACE codes authors are better to use names. For example CZ-NACE F is construction, so refer to it as construction.

Response 3

The titles of the figures and tables were changed following the recommendations provided. Codes NACE are used in tables and figures. The text is based mostly on names. The names are shorten many times and then the meaning does not need to be fully clear. The codes definy the branches the most appropriete.

Comment 4

Figure 1 should be named as „indicator values of the studied models for Healthy companies in Construction”. The x axis should be names as model number, y axis should be named as Z’score or whatever it is. The same holds for Figure 2. Figure 3 is not understandable in its current form. Instead of 25, 28 and F authors should have written CZ-NACE-25/28/F but the names are better here (Construction, Fabricated metals, Machinery and equipment). The figure should be named as „Absolute differences of the model indicators for Healthy companies regarding 2012 and 2017 by branches”. The same holds for Figure 4. The name of Table 4 should be changed to:

„The number of models within each branch confirming better conditions in expansion for Healthy companies„. The same is true for Table 5. In my humble opinion table 4 and 5 should be switched with a Figure and absolute frequency should be omitted and represent only the relative frequencies.

Response 4

The titles of the figures and tables were changed following the recommendations provided. The names of x and y axis were added. Codes 25, 28 and F were replaced by CZ-NACE 25, CZ-NACE 28 etc. The names were not selected because the shorten names do not have the full meaning.

Tables 4 and 5 were replaced by Figures 5 and 6.

Comment 5

The biggest mistake of the article is that it claims significant differences that are not proved and tested by statistical methods. For example in line 318: „in Figure 4. There are no significant differences between…”. I suggest the authors to run a non-parametric test for two related samples comparing model indicators per branches for years 2012 and 2017 like Wilcoxon test (Figure 1,2) or Friedman’s test for comparing model performances for the different branches (Figure 3,4).

Response 5

Testing by statistical methods was conducted. Results and interpretations are included in the text.

Firstly, Wilcoxon test has been added and commented in the text (table 3).

Secondly, the interpretations of Friedman’s test for comparing model performances for the different branches were added below Figure 4 (line 345 and further).

Round 2

Reviewer 1 Report

N/A

Reviewer 2 Report

The authors have diligently corrected the article and made the required changes. The study can now be accepted for publication. It was a good job.